# Brain Tissue-Derived Extracellular Vesicle Mediated Therapy in the Neonatal Ischemic Brain

**DOI:** 10.3390/ijms23020620

**Published:** 2022-01-06

**Authors:** Nam Phuong Nguyen, Hawley Helmbrecht, Ziming Ye, Tolulope Adebayo, Najma Hashi, My-Anh Doan, Elizabeth Nance

**Affiliations:** 1Molecular Engineering & Sciences Institute, University of Washington, Seattle, WA 98195, USA; nguyennp@uw.edu; 2Department of Chemical Engineering, University of Washington, Seattle, WA 98195, USA; hhelmbre@uw.edu (H.H.); zimingye@uw.edu (Z.Y.); nhashi@uw.edu (N.H.); 3Department of Biochemistry, University of Washington, Seattle, WA 98195, USA; 4Department of Biology, University of Washington, Seattle, WA 98195, USA; tolula@uw.edu; 5Department of Bioengineering, University of Washington, Seattle, WA 98195, USA; doanm@uw.edu

**Keywords:** extracellular vesicles, neonatal hypoxia ischemia, organotypic brain slice, oxygen glucose deprivation, IL-10, microglia morphology

## Abstract

Hypoxic-Ischemic Encephalopathy (HIE) in the brain is the leading cause of morbidity and mortality in neonates and can lead to irreparable tissue damage and cognition. Thus, investigating key mediators of the HI response to identify points of therapeutic intervention has significant clinical potential. Brain repair after HI requires highly coordinated injury responses mediated by cell-derived extracellular vesicles (EVs). Studies show that stem cell-derived EVs attenuate the injury response in ischemic models by releasing neuroprotective, neurogenic, and anti-inflammatory factors. In contrast to 2D cell cultures, we successfully isolated and characterized EVs from whole brain rat tissue (BEV) to study the therapeutic potential of endogenous EVs. We showed that BEVs decrease cytotoxicity in an ex vivo oxygen glucose deprivation (OGD) brain slice model of HI in a dose- and time-dependent manner. The minimum therapeutic dosage was determined to be 25 μg BEVs with a therapeutic application time window of 4–24 h post-injury. At this therapeutic dosage, BEV treatment increased anti-inflammatory cytokine expression. The morphology of microglia was also observed to shift from an amoeboid, inflammatory phenotype to a restorative, anti-inflammatory phenotype between 24–48 h of BEV exposure after OGD injury, indicating a shift in phenotype following BEV treatment. These results demonstrate the use of OWH brain slices to facilitate understanding of BEV activity and therapeutic potential in complex brain pathologies for treating neurological injury in neonates.

## 1. Introduction

Hypoxic-Ischemic Encephalopathy (HIE) is the leading cause of morbidity and mortality in neonates, to which there is no cure. The onset of HIE is caused by a hypoxia-ischemic (HI) event, which is characterized as a reduction of adequate blood flow to the brain that can often lead to stroke or permanent brain damage frequently exacerbated by delayed diagnosis [1,2,3]. Without oxygen and glucose being delivered to the developing brain from blood, the tissues experience a sequence of global immune responses and inflammation [4]. HIE has worldwide impact, affecting 1–8 neonates for every 1000 live births in developed countries, and as high as 26 neonates for every 1000 live births in low resource countries [4,5,6]. This condition can cause long-term neurological disabilities such as cerebral palsy, epilepsy, and cognitive impairment if left untreated. Neonates with HIE are usually symptomatic shortly after delivery with abnormalities in posture, muscle movement, cognitive response, and seizures [7]. Complex risk factors such as in utero infection, antepartum bleeding, uterine rupture, and maternal socioeconomic standing make HIE difficult to predict and diagnose at the time of injury [8].

Despite its seriousness, there are limited options for therapeutic intervention for HIE in the neonatal brain. Currently, inducing therapeutic hypothermia (TH) by cooling infants to 33.5–35 °C for 72 h is the only proven treatment for moderate or severe HIE. However, TH only offers a 15% absolute reduction in the risk of death and disability and is resource-intensive, thus not a feasible option for many developing countries [9,10]. As HIE poses a heavy health burden on neonates and their families, it is imperative to develop treatments targeting important players in acute HI pathogenesis.

Extracellular vesicles (EVs) are one such player of injury response that have gained widespread attention over recent years. EVs are cell derived biomolecules that span a large size range from 30 to 1000 nm, are released by every cell type, and contain a wide repertoire of biomolecules such as proteins, lipids, and nucleic acids [11,12]. Thus, EVs are categorized by their size and pathway of biogenesis: exosomes are small vesicles (30–100 nm) that originate from endosomal multivesicular bodies fusing with the plasma membrane; microvesicles (100–1000 nm) are shed directly from the plasma membrane; and apoptotic bodies (>1000 nm) bleb from the cell during apoptosis. After release, EVs are taken up by cells via most major pathways: phagocytosis, clathrin raft endocytosis, membrane fusion, or receptor-mediated endocytosis [13,14]. Though the mechanisms of cell-specific uptake of EVs remain unclear, EVs contain cell-specific cargo that are horizontally transferred to neighboring cells, acting as a mode of cellular communication and impacting cellular behavior [15,16].

Due to their ability to regulate the immune response, EVs have emerged as important players in injury response, neuronal development, and neuronal proliferation within the brain [2,3,14,17,18]. A specific advantage of EV therapy for brain injuries and disease is that they can cross the blood brain barrier (BBB) [17,19]. Both in vivo and in vitro studies confirmed that treatment with cell culture-derived EVs leads to increased neuronal recovery in adult stroke and traumatic brain injury models [20,21,22,23,24]. Xin et al. demonstrated that intravenous application of mesenchymal stem cell (MSC)-derived EVs in a stroke rat model improved neurological outcomes, angiogenesis, and neurogenesis [25]. The ability of EVs to intercellularly transport unique cargo indicates a strong potential for use as a therapeutic vehicle and/or diagnostic tool.

In this study, we used whole brain tissue-derived EVs (BEVs) to investigate the therapeutic activity of endogenous EVs in neonatal HIE. Ex vivo postnatal (P) 10 rat brain slices are exposed to BEVs following oxygen glucose deprivation (OGD) to model neonatal HIE conditions. We investigated the therapeutic potential of BEVs through dose- and time-dependency studies, evaluated changes in several gene expression markers after BEV administration, and quantified the cellular immune response by performing morphological analysis of microglia, a key cell involved in injury and immune response in the brain [26]. From a clinical perspective, studying the brain’s response to endogenous BEVs during ischemic attenuation can provide a first step to understanding how therapeutic EV payloads can improve outcomes for neonatal HIE.

## 2. Results

### 2.1. Characterizing BEVs Isolated from Whole Neonatal Brain Tissue

Recent papers suggest that while several different techniques exist for EV sequestration, such as ultracentrifugation (UC), size exclusion chromatography (SEC), and density gradient (DG), a combination of two or more of these methods improve EV throughput [27,28,29]. BEVs from saline-perfused P10 male rat brain were isolated using a combination of UC, SEC, and ultrafiltration (UF). The size and number of BEVs were characterized using nanoparticle tracking analysis (NTA). NTA results for BEVs showed a mean size distribution of 100–300 nm (Figure 1A, Appendix A)—representative of the size regime of exosomes and small microvesicles—and a concentration that averaged 3.2 × 10^11^ particles/mL among formulations (Figure 1B). To put this value into perspective, blood has an estimated concentration of about 5–15 × 10^8^ particles/mL [30]. NTA results reported a purity similar to published tissue-derived EV values, confirming that our methods were appropriate for BEV purification [27]. The zeta potential of BEVs was −14.1 mV (average). We visualized the geometry of isolated BEVs using transmission electron microscopy (TEM) with negative and positive staining (Figure 1C, Appendix A). The negative stain coats the background with heavy metal, allowing electrons to travel more easily through the vesicle. This created high contrast between the background and clearly delineated the lipid bilayer of the BEVs. Meanwhile the positive stain coats organic material lining the surface of BEVs, which revealed surface texture likely due to membrane biomolecules such as lipids and proteins, in addition to the lipid bilayer. Together, these images provided information about the vesicular geometry and topology of BEVs. 

Dot blot immunodetection confirmed the presence of target proteins CD9, CD63, and GM130, and housekeeping protein GAPDH within BEV and brain tissue lysates following 1X radioimmunoprecipitation buffer (RIPA) lysing. The 1X PBS control produced no fluorescence, while the brain tissue lysates (for all dilution factors) showed strong fluorescent signals for all targeted proteins (Figure 2). Using ImageJ, we quantified the total detected fluorescent signal and computed the integral signal density values within a selected region of interest to serve as a proxy for the total amount of protein assessed in the sample (Appendix A). Detection of EV positive markers, tetraspanins CD9 and CD63, were confirmed in all blots containing BEV and brain lysate. As expected, the BEV lysate showed negligible signal for negative EV biomarker GM130 (0.004 signal density), confirming that BEV samples were free of cellular contaminants otherwise found in brain tissue [31]. From integral signal density quantification of the BEV lysates, we determined that CD9 had the greatest abundance of all measured proteins (0.339), followed by CD63 (0.302) and GAPDH (0.044). A signal density ratio quantifies the abundance of target proteins extracted in the BEV lysate compared to brain tissue lysate (signal density ratio = signal density of BEV/signal density of brain tissue lysate). The signal density ratios for CD9, CD63, GM130, and GAPDH are 69.5%, 64%, 2.12%, and 12.19%. Based on these immunoblotting results, we validated the identity of EV isolates used for these studies, with CD9 being the most abundant target tetraspanin detected.

### 2.2. BEV-Mediated Therapeutic Effects on an Ex Vivo Ischemic Slice Model

To study EV therapeutic activity and its impact on cellular behavior, we performed oxygen glucose deprivation (OGD) in organotypic whole hemisphere (OWH) slices from the P10 rat brain to model HI in term-equivalent neonates [32,33]. OWH brain slices uniquely capture the 3D cytoarchitecture and regional complexity of the brain [33,34]. OWH slices allow for the simultaneous assessment of different brain regions, such as the cortex, hippocampus, thalamus, and others, which are not present collectively in standard organoid, cortical, or hippocampal culture models [35]. Following OGD conditioning, we topically applied 5 μg, 12.5 μg, 25 μg, and 50 μg BEV dosages on slices at various timepoints (Figure 3A). Healthy and OGD control slices did not receive any BEV treatment. Comparing cellular cytotoxicity values among ex vivo slices revealed several important trends. First, the percent cytotoxicity expectedly decreased over time in both the healthy and OGD control conditions as tissues recovered from acute slicing (Figure 3B) [33]. Second, 25 μg is the observed minimum therapeutic dosage beginning at 24 h exposure. At this exposure time, the percent cytotoxicity was not statistically significant from the healthy control but was statistically significant from the OGD (0 μg BEV) control (*p* < 0.0001). This reveals that 24 h is the observed minimum exposure time necessary to elicit a therapeutic response at the minimum therapeutic dosage. Third, the percent cytotoxicities across 5–50 μg dosages were statistically insignificant from the healthy control at 48 h exposure time (*p* > 0.999), which is regarded as the therapeutic exposure time.

After determining 25 μg and 24 h as the therapeutic dose and exposure time, we next sought to establish a therapeutic window for BEV application. Experiments were performed on ex vivo slices following BEV treatment at various application timepoints before and after OGD conditioning (−24, 4, 24 h). Noticeably, 5–50 μg dosage groups elicited therapeutic effects resulting in cell viability at similar levels to the healthy control (Figure 3C). This result occurred in BEV application as early as 4 h post-OGD and as late as 24 h post-OGD and indicates that the therapeutic window of administration can extend up to a day post-injury. The results from this experiment suggest that BEV application is therapeutically effective even when administered several hours after injury onset.

### 2.3. RNA Expression Changes in Response to BEV Treatment

Though brain slices exposed to BEVs demonstrated therapeutic effects, it remains unclear what cytokines and regulatory factors are involved. In addition to assessing cell viability via confocal imaging, RT-qPCR was used to further quantify BEV therapeutic activity on ex vivo OGD slices. RT-qPCR was run on a subgroup of slices to evaluate temporal changes in the gene expression profiles of inflammatory (IL-4, IL-6, IL-9, IL-10, IL-11, N κβ), oxidative stress (iNOS), cell death (Casp-3), and cellular activation markers (Ki67, CD68, GFAP, Synapsin, CD11b, Vim) (Table 1) following BEV treatment at 25 μg for 24 h (Figure 4). At 24 h, slices with BEV treatment showed a significant increase (*p* = 0.037) in the expression of IL-10 compared to the OGD control (Figure 4A). After 24 h, there were no significant changes in expression of cell activation and cell death markers profiled in this study (Figure 4B,C).

### 2.4. Morphological Response of Glial Cells to BEV Treatment

Features of microglia were compared across healthy control, OGD control, and BEV treated slices at varying exposure times—4 h, 24 h, and 48 h (Figure 5). Qualitative differences of density and morphology were observed in representative images of microglia stained with Iba-1 across the healthy, OGD, and BEV-treated slices at 24 h (Figure 5A, Appendix A). The healthy control slices showed the highest microglia density, followed by BEV-treated slices, and finally OGD control slices with the lowest density. Microglia from healthy and BEV-treated slices also displayed greater extent of ramification than microglia from OGD control slices. Quantitative analysis of microglia shape features was completed with Python-based image processing (Figure 5B–F). While fluctuations in morphology during the first 24 h may be due to natural microglial heterogeneity, changes in perimeter, area, circularity, and aspect ratio were observed in BEV-treated slices between the 24-h and 48-h exposure times (Figure 5B–E). At the 24-h time point, the microglia from BEV-treated slices showed no significant difference from the OGD control slices in perimeter, area, circularity, and aspect ratio. However, this trend changes at the 48-h time point as the microglia from BEV-treated groups become statistically differentiable to the healthy and OGD controls in every feature except perimeter (area: *p* < 0.0001 for both healthy and OGD controls; circularity: *p* < 0.0001 for both healthy and OGD controls; aspect ratio: *p* = 0.0002 for healthy control, *p* = 0.0097 for OGD control). Specifically, the median area and circularity of the BEV-treated microglial cells decreased while the median aspect ratio and perimeter increased; the OGD control slices observed increases in the median value of circularity, area, perimeter and observed relatively no change in median aspect ratio; the healthy control slices observed an increase in median value for circularity and a decrease in the median value for area, perimeter, and aspect ratio. Fold changes between the healthy control and both the OGD control and BEV-treated slices were visualized via individual 2-color heat maps for the perimeter, area, circularity, and aspect ratio of the microglia (Figure 5F, Appendix A). The heatmaps visually showed that at 24 h the fold change between microglia from OGD control slices, BEV-treated slices, and healthy controls were similar across all geometric features. Meanwhile, the fold change at 48 h showed visually different intensities between the microglia of the OGD control and BEV-treated slices in every feature except perimeter, with a more negative fold change for perimeter, area, and circularity and a more positive fold change in aspect ratio for the microglia in BEV-treated OGD slices.

Microglia were split into five distinct shape modes (SM) (Figure 6) and graphed for perimeter, area, circularity, and aspect ratio (Figure 6A). Representative images of the original Iba-1 stain, segmented microglia, and color-coded SMs are displayed alongside a representative dendrogram of the PCA and k-means clustering determined SMs (Figure 6B). Based on the dendrogram and representative image, SM3 represents a circular microglial shape with the highest circularity yet smallest perimeter, area, and aspect ratio. SM1 and SM5 were statistically indifferentiable amongst the four geometric features analyzed and correlated with branched microglia with little to no swelling, as these SMs have the highest perimeter and area and the lowest circularity. SM2 and SM4 were nearly indistinguishable in the three of the geometric features, but SM4 has a significantly higher aspect ratio. The aspect ratio differences between SM2 and SM4 may indicate that SM4 has increased process extension or unidirectional branching in comparison to SM2. Since SMs 2 and 4 include geometric values mainly between the SM1/SM5 set and SM3, the geometric features support slightly branched microglia with increased swelling. Frequencies of all SMs were visualized as a heat map across every exposure time—0, 4, 24, and 48 h—and for all three experimental groups—healthy control, OGD control, and BEV treatment (Figure 6C). SM3 and SM4 exhibited the highest overall frequency with the peak of SM3 occurring at 24 h and the peak of SM4 occurring at 0 and 4 h. SM 1 and 5 exhibited the lowest frequencies with the lowest overall frequencies occurring in SM 5 at 24 h. The BEV-treated slices showed the lowest microglial heterogeneity at 24 h when SM3 dominated at 37% and both SM1 and SM5 were at 12%. However, at 48 h the BEV-treated slices showed the highest microglial heterogeneity with the most evenly dispersed frequencies across all groups and exposure times.

## 3. Discussion

EVs initiate and promote a therapeutic injury response in adult stroke models by releasing neuroprotective, neurogenic, and anti-inflammatory factors and have high potential to be translated into the neonatal therapeutic space [36,37,38]. However, most studies and all ongoing clinical trials evaluating EV therapeutic potential are performed on adult models using stem cell-culture derived EVs, which do not recapitulate EV activity in their endogenous environment [21,22,37,39]. Here, we investigated the therapeutic potential of BEV administration on an OWH slice culture model of neonatal HI. We showed that BEVs can be successfully isolated from whole brain tissues by using a combination of UC, SEC, and UF techniques. Independently, these are standard techniques in EV research but using them in combination allowed for increased EV yield and purity as quantified by NTA and BCA. We validated the identity of BEV using dot blots targeting positive (tetraspanins CD9, CD63) and negative (GM130-a membrane protein of the Golgi Apparatus) EV markers compared to brain tissue lysate. Positive markers were detected in both the BEV and brain tissue lysates, while negative EV markers were only present in brain tissue lysate confirming the identity of isolated BEVs. Both the negative control (1X PBS) and negative EV marker GM130 had negligible signal density values, which is indicative of BEV purity. It is difficult to address the purity of the BEV samples, as quantitative standards have not been set due to inconsistent methods utilized between research groups for EV isolations. This problem is compounded by the diverse sources from which EVs are extracted. The wide range of EV sequestration methods result in highly heterogenous distributions of EV size across the literature. We report our BEV isolation data, including size distribution, and concentration to provide comprehensive characterization of our BEV isolates (Appendix A).

We performed dose- and time-dependency experiments to evaluate the therapeutic potential of BEVs in a HI brain slice model. We observed an expected decrease in the percent cellular cytotoxicity over time in the healthy and OGD control conditions as the slices naturally recover from acute slicing, aligning with previous outcomes in our OGD slice model in rats [33]. Results from these studies suggest that 25 μg of BEVs is the minimum therapeutic dosage after 24 h exposure time. At this dosage and exposure time, BEV-treated slices and the healthy control display similar cytotoxicity values. Notably, at 4 h of exposure time all BEV dosages elicited an increased cytotoxicity compared to the OGD control. However, this increase was not observed at 24 h. We hypothesize this may be due to a delayed activation of immune response following injury onset that occurs between 4 and 24 h [40]. Future work could investigate the temporal change in therapeutic activity of BEVs post-OGD. Additionally, 48 h is considered as the therapeutic exposure time for which cellular cytotoxicity significantly decreased following BEV treatment across all tested dosages.

We sought to determine the therapeutic application window of BEV treatment post-OGD conditioning. Cellular cytotoxicity values of BEV treatment were comparable to the healthy control at all dosages when applied between 4 and 24 h post-OGD conditioning. This indicates that the application window of BEVs spans at least 24 h following injury onset. In our interpretation, this heightened therapeutic activity suggests that BEV treatment may directly interact with important cells regulating injury response during peak immune activation period within 24 h post-injury [40]. Unexpectedly, 24 h BEV priming of the ex vivo slices before OGD (−24 h) in our time-dependent experiments (Figure 4C) did not demonstrate increased therapeutic response. It may be that priming will be more efficient closer to the period of injury onset, or that priming requires cell-specific BEVs to trigger neuroprotective factors that will lead to a decrease in cell cytotoxicity. To reduce the number of animals used, we did not run 50 μg dosage experiments for the application-time dependent experiments because 25 μg was previously determined as the therapeutic dosage. Dosages greater than 25 μg could continue to elicit observed therapeutic trends, but this may not be necessary as 25 μg drove cytotoxicity values down to levels comparable to the healthy control after 24 h (Figure 4B). Furthermore, both BEVs and OWH slice studies were done using male pups; clinically, males often have worse outcomes in neurological injury than females [41]. While 25 μg BEV is presented as the minimum therapeutic dosage for ischemic male rat pups, this dose may be lower for a pup with a lesser injury severity regardless of sex. Future studies investigating the role of sex-dependent BEV injury attenuation will provide insight into the biological conditions affecting BEV therapeutic efficacy.

While our cytotoxicity evaluations did indeed demonstrate reduced cytotoxicity in the BEV-treated slices, we evaluated qPCR on two cell death markers: iNOS and Casp-3. qPCR results indicated no difference in these specific cell death markers; therefore, it is likely to be the case that BEVs impact another pathway not explored in this paper. Other studies have found that EVs influence other apoptotic markers such as BCL2 Associated X (Bax) [23], superoxide dismutase 1 (SOD-1) [24], Bcl-2 and nineteen kilodalton-interacting protein 2 (BNIPS) [42] and osteopontin (OPN) [43]. It is also possible that rather than influencing cell death pathways directly, BEVs impact the expression of cell survival pathways such as extracellular signal-regulated kinase 1 (Erk1) and Erk2 [24] or prion protein (PrP) [44]. These mechanisms of effect will need to be explored in future studies.

Results from whole slice qPCR analysis also demonstrated a significant increase in anti-inflammatory gene expression for IL-10 in BEV-treated slices compared to the OGD control group after 24 h exposure. In particular, IL-10 expression after BEV treatment is comparable to the healthy control, pointing to the role that BEVs play in regulating injury attenuation. IL-10 is an anti-inflammatory cytokine, and the upregulation of IL-10 mRNA expression suggests that BEVs play a role in alleviating inflammation in injured tissues through elevating IL-10 anti-inflammatory expression. As changes in microglial phenotypes are connected to inflammation, disease onset and progression, and stimuli from the local environment [45], we paired the gene expression analysis with phenotypic profiling of microglia to explore the role that BEVs play in an HI model. Microglia differentiation between BEV treatment, OGD controls, and healthy controls was quantified over several key morphologic features on both an individual and population-based level. On the individual level, microglia in BEV-treated groups demonstrate unique morphological changes between 4–48 h of BEV exposure. At 4 h, microglia in both the BEV treated slices and OGD slices show significant increases in circularity from the healthy control slices, suggesting swelling of microglia. Swollen, thicker microglia are correlated with a pro-inflammatory state, which is confirmed by increased cytotoxicity and pro-inflammatory marker expression at 4 h of exposure [46]. Microglia from BEV-treated slices start to exhibit significant morphological changes from OGD controls at 48 h. The microglia from BEV exposure at 48 h displayed decreased median area and circularity with an increased aspect ratio, which suggests that the microglia are entering more elongated shapes. Elongation with decreased area also indicates a decrease in swelling associated with a less inflammatory microglial state. The morphological shift between 24–48 h suggests a change in microglial activity at the previously reported crucial timepoint in ischemic intervention.

This morphological shift between 24–48 h is also observed at the population-level in microglia exposed to BEV treatment. Microglia are morphologically heterogeneous even in a healthy state and exhibit functional diversity [47]. Using the SMs obtained via VAMPIRE, overall morphological changes in the microglial population were compared across treatment groups and exposure times. BEV-treated slices displayed increased homogeneity of microglial phenotypes with the highest increase in SM3—whose SM is correlated with circular, ameboid microglia—at 4 h and 24 h; microglia from OGD control slices showed increasing homogeneity from 4 h to 24 h with SM4 the highest at 4-h exposure and SM 3 the highest at 24 h. The similar increases between the OGD and BEV-treated slices supports microglial transition to an inflammatory state up to the 24-h time point. However, at 48 h, while the OGD and healthy control homogeneity increases (with SM3 remaining the highest frequency), microglia from BEV-treated slices become more heterogeneous and transition from the highest frequency SM to SM1—a higher branched and less swollen SM. Strikingly, the increase in heterogeneity and increased IL-10 marker from qPCR analysis in BEV-treated slices suggests that the microglia not only transition out of a pro-inflammatory state between 24 and 48 h, but also enter into a restorative phenotype rather than a resting state characteristic of the healthy control. The changes in microglial morphology reveal that BEV administration has dose- and time-dependent impacts upon immune cell activation within an injured neonatal brain. Though no significant changes in cell activation or cell death marker expression were observed at 24 h of exposure, it is hypothesized that these changes will occur at 48 h to reflect the time-dependent shift observed in microglia morphology. Future studies evaluating chronologic changes in gene expression will be valuable in bridging changes in microglial morphology with cellular response in an HI model following BEV administration.

Multiple P10 rat brains are required for each BEV extraction sample, so tissues from pups of the same litter and sex were pooled together to reduce biological variability amongst samples. Despite our efforts to reduce biological variability, there may still be discrepancies between pups from different litters that affect our results. For example, pups that are malnourished are likely to have under-developed brains that may impact the severity of immune response to injury. The small mass of neonatal brains in animal models poses a limitation in EV research requiring large sample quantities. It should be noted that when considering the translation of this work into a clinical setting, it is unlikely that BEVs will be derived from the neonatal brain and administered into patients. Rather, the goal of this research is to first investigate the therapeutic potential of BEVs in ischemic attenuation, so that future directions of our work can focus on teasing apart specific aspects of protection against injury. Clinical translation of this work will involve comparing the protective mechanisms and payloads imparted by BEVs to conventional forms of cell culture-derived EVs, such as mesenchymal stem cells. Understanding endogenous BEV activity in a neonatal ischemic model will inform the design of EV therapeutics from the bench into clinical use. While we demonstrated that BEVs elicit dose- and time-dependent therapeutic effects when applied to OGD conditioned slices, endogenous EVs exist in a heterogeneous population within the brain. Therefore, all results reported in this paper are indicative of treatment from the BEV population represented by EVs derived from different cell types. Recent studies suggest that different cell specific EVs play different roles within the brain. For example, astrocyte-derived EVs are reported to be heavily populated in stroke in vivo models and regulate the release of neurotrophins to injured tissues [42,48]. This motivates further exploration of the role and mechanism of effect for cell-specific EVs in the pathogenesis of HIE.

## 4. Materials and Methods

### 4.1. Animal Care and Ethics

This study was performed in accordance with the guide for the care and use of laboratory animals of the National Institutes of Health (NIH). All animals were handled according to an approved Institutional Animal Care and Use Committee (IACUC) protocol (#4383-02) of the University of Washington (UW), Seattle, WA. The UW has an approved Animal Welfare Assurance (#A3464-01) on file with the NIH Office of Laboratory Animal Welfare, is registered with the United States Department of Agriculture (certificate #91-R-0001), and is accredited by AAALAC International. Time-mated pregnant female Sprague–Dawley rats (virus antibody-free CD^®^ (SD) IGS, Charles River Laboratories, Raleigh, NC, USA) were purchased and arrived on postnatal day 5 with a litter of 10, sex-balanced pups. Dams were housed individually with their litter and allowed to acclimate to their environment. Before and after the experiment, each dam and her pups were housed under standard conditions with an automatic 12 h light/dark cycle, a temperature range of 20–26 °C, and access to standard chow and autoclaved tap water ad libitum. The pups were checked for health daily.

### 4.2. OWH Slicing Methodology 

Following intraperitoneal euthanasia with pentobarbitol, fresh brain tissue was rapidly extracted from P10 male rats, placed in ice cold dissection media, and sectioned into 300 μm slices using a Mcllwain tissue chopper (Ted Pella, Redding, CA, USA). These slices were plated onto 30 mm cell culture inserts (Millipore Sigma, Burlington, MA, USA) in nontreated 6-well plates (USA Scientific, Orlando, FL, USA). Slices were then incubated at 37 °C in 1 mL of 5% slice culture medium (SCM) at 37 °C and 5% CO_2_ to recover from acute slicing. SCM medium with 5% horse serum consists of 180 mL HBSS (Gibco), 20 mL horse serum, 200 mL MEM (Gibco, Dublin, Ireland), 4 mL PenStrep, and 4 mL GlutaMax (Gibco). Days in vitro (DIV) 0 was defined as the day of brain extraction and slicing. At DIV1, the media was exchanged with fresh 5% SCM.

### 4.3. OGD Methodology 

Acute brain slices are incubated in oxygen-glucose deprived media within an oxygen-free chamber [33]. Since multiple slices can be obtained with one brain, OWH slices reduce the number of animals required per experiment and allow for higher throughput screening. At DIV3 slices were placed in glucose depleted media and a hypoxic chamber for 30 min before further tissue processing [33]. OGD medium consists of 120 mM NaCl, 5 mM KCl, 1.25 M NaH_2_PO_4_, 2 mM MgSO_4_, 2 mM CaCl_2_, 25 mM NaHCO_3_ and 20 mM HEPES. The solution was passed through a vacuum filter unit 0.2 μm (Nalgene, Rochester, NY, USA). Previous work in the rat determined that 30 min incubation time under these conditions was sufficient to induce significantly higher cell death compared to healthy slices [33]. Varying OGD exposure can result in a high degree of cell death and depletion of glutathione (GSH), a mediator of intracellular oxidative stress [33]. 

### 4.4. BEV Isolation Using a Combination of Methods

A modified BEV isolation procedure published by Vella et. al. was used to extract and enrich BEVs from whole rat brain [49]. Each rat brain was perfused with 10 mL PBS to reduce contamination from blood and serum-derived EVs. For every extraction, four PBS perfused P10 brains were used from the same litter and sex to limit biological variability and flash frozen prior to processing. Whole brains extracted from P10 rats were finely chopped in a solution and incubated for 20 min in a water bath with protease inhibitors to allow for complete dissociation of extracellular matrix proteins. Subsequently, tissue samples were ultracentrifuged under increasing speeds at 4 °C to remove large proteins and cellular debris from the supernatant. The homogenate was initially spun at 300× *g* for 5 min, transferred to a new tube to spin at 2000× *g* for 10 min, and finally transferred to new tubes and ultracentrifuged for 10,000× *g* for 35 min. After ultracentrifugation, the collected supernatant was run through an Amicon ultrafiltration column (100 kDa molecular weight cutoff) and spun at 3214× *g* for 90–120 min, or until the final volume reached 500 μL. A size exclusion chromatography column (iZon Science, Portland, OR, USA) was then used to further purify BEVs, and fractions containing high concentrations of BEVs were ultracentrifuged in an Amicon ultrafiltration column (50 kDa) at 3214× *g* for 60–90 min at 4 °C to concentrate isolated BEVs [50]. The concentrate was collected and flash frozen at −80 °C in 1X PBS until use for experiments [50].

### 4.5. BEV Characterization

The total protein concentration of the isolated BEV solution was quantified using a bicinchoninic acid assay (BCA, Pierce BCA Protein Kit). Nanoparticle tracking analysis (NTA) using a NanoSight was used to quantify particle size (nm) at a dilution of 1:1000 in 1X PBS. The purity of BEVs was calculated as the number of particles measured by NTA/protein concentration measured by BCA [51]. The zeta potential of BEVs suspended in 1X PBS was measured using a Zetasizer (Malvern Panalytical, Malvern, UK).

The identity of isolated EVs was verified using dot blot immunodetection. The dot blots were labeled for positive and negative surface markers. Positive EV markers included: tetraspanins CD9 (1:500, BD Biosciences, Franklin Lakes, NJ, USA) and CD63 (1:500, BioRad, Hercules, CA, USA). The negative EV protein marker used is a membrane protein of the Golgi Apparatus GM130 (1:500, BD Biosciences, Franklin Lakes, NJ, USA). A housekeeping marker GAPDH (1:500, Thermofisher, Waltham, MA, USA) was also used as a positive control for the tissue lysates, while 1X PBS was used as the negative control. BEVs and P10 brain tissue were lysed using 1X RIPA buffer with a 1X protease inhibitor cocktail on ice (100x, Thermofisher, Waltham, MA, USA). Lysates were incubated on ice for 20–30 min following the addition of 1X RIPA. The brain tissue lysates were ultracentrifuged at 15,000× *g* for 30 min at 4 °C to pellet cell debris, which was then removed for dot blotting. BEV lysates were concentrated using a 3 kDa molecular weight cutoff ultrafilter (Thermofisher, Waltham, MA, USA) and ultracentrifuged at 15,000× *g*. A BCA quantification was used immediately prior to dot blotting to determine the protein mass loaded onto each membrane. Strips of wetted polyvinylidene fluoride (PVDF) transfer membranes were prepared (Thermofisher, Waltham, MA, USA), and a hydrophobic pen was used to draw rings for each sample protein. Proteins from BEV and brain tissue lysate were aliquoted at a volume of 15 μL into each ring. Following an initial blocking with 5% skim milk in TBST buffer at room temperature, the membranes were blocked with primary antibodies diluted with 5% skim milk TBST buffer on a shaker overnight at 4 °C. The following morning, the membranes were blocked with secondary antibody (IR800 goat anti-mouse, 1:4000) for 1.5 h in the dark at room temperature. After staining with secondary antibody, the membranes were washed with 5% skim milk and TBST buffers several times before imaging (Azure Biosystems, Dublin, CA, USA).

Dot immunoblots were analyzed using ImageJ and the integral signal density (total fluorescent signal) of each sample was quantified within a region of interest of the same area for each sample. All sample fluorescent intensities were normalized with the blank solution (PBS) prior to quantification.

### 4.6. TEM Imaging

TEM Imaging was performed using a FEI (Thermoscientific) TF20 TEM at the UW Molecular Analysis Facility at 200 kV with an Eagle CCD Camera. To prepare BEV samples on a carbon grid, aliquots of BEVs were thawed and mixed with an equal volume of 1.5% glutaraldehyde buffered with 0.1 sodium cacodylate buffer for fixing [50]. Then, 200 mesh TEM grids were glow-discharged, and 10 μL of BEV suspension was placed onto the grid and left to dry for 5 min in ambient temperature. The grids were then exposed to 5% uranyl acetate for 5 min and washed three times with a droplet of distilled water. Grids were air-dried before storage.

### 4.7. BEV Administration on Ex Vivo Slices

Frozen BEV fractions were thawed and gently resuspended with a pipette. BEV solution was then diluted with sterile 1X PBS to achieve experimental dosages (5 μg, 12.5 μg, 25 μg, 50 μg) suspended in a total volume of 100 μL per slice. The BEV solution was topically applied onto the tissue slices at various application timepoints (−24, 0, 4, and 24 h post-OGD conditioning), aligned with previous therapeutic efficacy studies performed in OWH slices [33,34]. After various BEV exposure timepoints of interest (4, 24, and 48 h), slices were stained with propidium iodide (PI) to quantify cell viability and fixed with 4% formalin. Formalin fixed slices were subsequently co-stained with DAPI and Iba-1 (Wako), a standard marker of microglia.

### 4.8. Confocal Imaging

A Nikon A1R confocal microscope was used for blinded quantification of cell viability (PI-stained cell count/ total cell count). Three to five images were captured of the cortical region of every slice at 40× magnification and each image was blinded to participants involved in cell counting with ImageJ. Confocal images of cells were taken at 40× magnification and further processed computationally.

### 4.9. Cell Morphology Analysis

All images of interest from each treatment and control group of Iba-1-stained slices were converted from .nd2 file format to .tiff file format using the Nikon’s Confocal NIS-Elements software. The saved .tiff files of all images were segmented by multiple thresholding methods using scikit-image’s try_all_threshold functionality which includes the isodata, li, mean, minimum, otsu, triangle, and yen thresholds [52]. An image showing the original cell image compared with the other seven thresholding methods was saved as a .tiff for qualitative comparison. The Li threshold method for segmentation was determined to be most accurate when comparing to manual segmentation with the Fiji implementation of ImageJ2 [53]. For further quantification, the images were individually segmented using scikit-images threshold_li function followed by removing small objects smaller than 71 pixels^2^ and filling all holes. To determine the size of the small object to remove, we accounted for the average size of microglia (1600 μm^2^) [54]. The microglia size was converted to a pixel cut off by (1) converting from pixels to μm with the confocal metadata 1 pixel = 3.4527 um conversion (2) ensuring no potential microglia were cut off by selecting a lower boundary at half the average microglia size.

After segmentation, cells were quantified with two methods: regionprops analysis through Sci-kit image and with Visually Aided Morpho-Phenotyping Image Recognition (VAMIRE) [55]. With the regionprops functionality, cells were individually measured for geometric parameters including area, perimeter, major axis length, and minor axis length. The aspect ratio of cells was calculated as the ratio of the major axis length to the minor axis length and cellular circularity was calculated with the following equation:$$Circularity=\frac{4\pi \;*\;Area}{Perimeter^{2}}$$

Heat maps of the fold change of individual geometric features from regionprops analysis were created using Excel conditional formatting with a custom 2-scale color map with white as the highest value and a unique color as the lowest value—perimeter: green, area: yellow, circularity: blue, and aspect ratio: purple.

For VAMPIRE analysis, the Numpy files of the cell images after threshold-based segmentation were split into four equal quadrants. For each quadrant, any cell touching the edge of the image was removed to avoid noise created by artificial straight edges. All the quadrants were split into groups for building and applying a VAMPIRE model via an 80:20 train:test split. Models were built and applied according to previously published VAMPIRE methodology [55]. Five shape modes (SM) where chosen to capture biological variation while remaining computationally efficient. The VAMPIRE method also produced results of the aspect ratio, circularity, perimeter, and area of every cell and correlated shape mode. Qualitative visualizations of the segmented cells were created with Python by assigning each shape mode a color with the Matplotlib twilight color map [56]. Heat maps of shape mode frequencies were created in Microsoft Excel with conditional formatting and a red-yellow-green color scale. All code can be found at: https://github.com/Nance-Lab/cellmorphflows/tree/master/Phuong_BeV%20, accessed on 22 December 2021 [57].

### 4.10. Reverse Transcriptase Quantitative Polymerase Chain Reaction (RT-qPCR)

Changes in pro-inflammatory, anti-inflammatory, cell death, glial activation, and oxidative stress gene expression profiles following BEV treatment was quantified using RT-qPCR. For each condition, three brain slices were cultured together on the same membrane. Slices were preserved in RNALater (Thermofisher, Waltham, MA, USA) and kept at 4 °C prior to processing to prevent RNA degradation. The RNA from homogenized brain slices were extracted with TRIzol reagent, pelleted at 15,000× *g*, washed several times with ultrapure DEPC water (Thermofisher, Waltham, MA, USA) and 70% ethanol, and the RNA final concentration was measured using a NanoDrop. cDNA was diluted to 20 ng/μL with ultrapure RNA-free water. RNA was transcribed into cDNA using Thermofisher (Waltham, MA, USA) Reverse Transcription RNA to cDNA kit. qPCR was run using the transcribed cDNA and BioRad (Hercules, CA, USA) SYBR Green Master Mix that binds to double-stranded DNA to quantitatively track the progress of DNA amplification in real-time. Primers used were: Inflammatory (IL-4, IL-6, IL-9, IL-10, IL-11, Nκβ), oxidative stress (iNOS), cell death (Casp-3), and cellular activation markers (Ki67, CD68, GFAP, Synapsin, CD11b, Vim), and a housekeeping gene (GAPDH) (Table 1) [58]. The qPCR runs at 95 °C for 30 s, 95 °C for 5 s, and then 60 °C for 30 s for 40 cycles. The gene expression changes in OGD-conditioned and BEV treatment groups were normalized to the healthy controls to quantify fold-expression change. Results were statistically analyzed with one-way ANOVA (Kruskal–Wallis multiple comparisons tests) and normalized by the median Cq value for all samples.

### 4.11. Statistical Analysis

Statistical analysis was performed in GraphPad Prism version 9.2.0 (GraphPad Software, San Diego, CA, USA). Graphs for the cell features compared to the control and treatment groups and compared to shape modes are displayed as the median with interquartile range and all data points are shown. Morphology features were compared across all groups using a nonparametric One-Way ANOVA utilizing the Kruskal–Wallis test and Dunn’s post hoc correction for multiple comparisons. All *p*-values < 0.05 were considered statistically significant.

## 5. Conclusions

EVs have gained much interest due to their emergence as important players in injury response, neuronal development, and proliferation within the adult brain. We investigated the dose- and time-dependency of BEV treatment on an ex vivo slice model of HI. In contrast to existing studies, we performed experiments using whole brain tissue derived EVs to recapitulate the therapeutic potential of endogenous EVs on neonatal ischemic models. In summary, we showed that BEVs can be successfully derived from whole neonatal rat brain tissue through a combination of EV isolation and purification techniques. BEV treatment of ex vivo HI slice models decreased cellular cytotoxicity at a minimum therapeutic dose of 25 μg and a therapeutic exposure time of 48 h, and has an application time window up to 24 h post-injury. We observed a shift in microglial morphology from pro-inflammatory amoebic to anti-inflammatory and restorative shape modes following BEV administration. Analysis of changes in gene expression profiles indicated that BEVs significantly increased anti-inflammatory IL-10 cytokine expression. Collectively, our results reflect the promising therapeutic role that BEVs play in attenuating inflammation and cell death in HIE neonatal models. Future research in BEVs can inform the design and administration of therapies for improving outcomes for neonatal HIE patients.

## Figures and Tables

**Figure 1 ijms-23-00620-f001:**
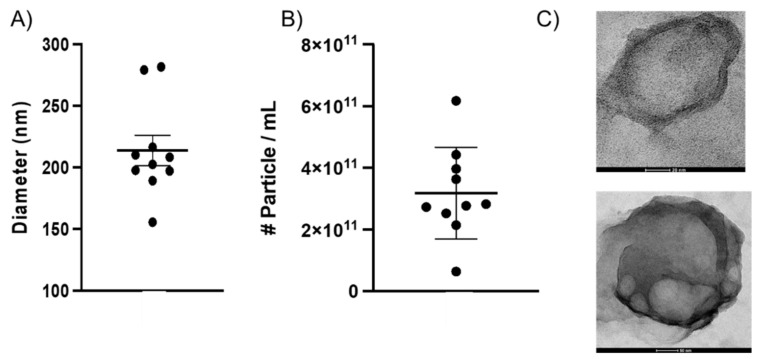
Average (**A**) diameter and (**B**) concentrations (#particle/mL) across all BEV extracts as measured by NTA and BCA. Each point represents one unique sample (*N* = 10). The mean diameter and concentration across all isolates are 213.9 nm and 3.2 × 10^11^ particles/mL, respectively. (**C**) High magnification transmission electron microscopy (TEM) of isolated BEVs using negative staining (**top**) and positive staining (**bottom**) with visible lipid membrane structure. BEVs ranged from 30 to 200 nm, supporting NTA data. Scale bars = 20 nm (**top**), 50 nm (**bottom**).

**Figure 2 ijms-23-00620-f002:**
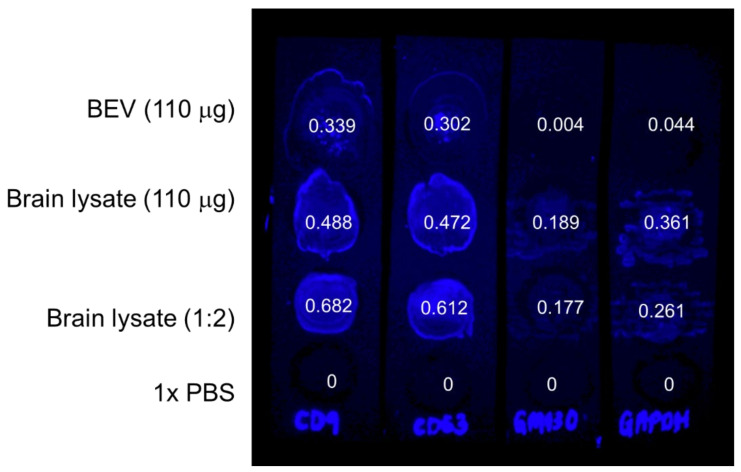
Dot blot immunodetection of BEV lysates compared to brain tissue (BT) lysate and PBS. Samples (top to bottom) are BEV lysate, BT lysate, BT lysate (1:2 dilution in PBS), and PBS (negative control). Each membrane strip was probed with a different primary antibody for a specific target protein (left to right): CD9, CD63, GM130, and GAPDH. Each spot contained 110 μg (or 55 μg) of sample protein in a volume of 15 μL as measured by BCA. The total fluorescent signal density of each dot was normalized by the negative control (PBS), quantified in ImageJ, and labeled in the corresponding dot.

**Figure 3 ijms-23-00620-f003:**
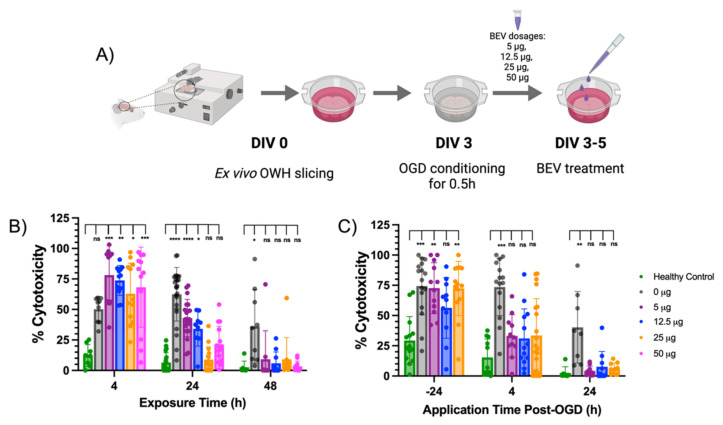
Dose- and time-dependent evaluation of cell viability after bEV treatment on ischemic brain slices. (**A**) Timeline of ex vivo OWH slicing, OGD conditioning, and BEV treatment prior to evaluating cellular cytotoxicity. (**B**) Slices were immediately exposed to BEVs following OGD treatment for 4, 24, and 48 h. Dosages of 25 μg (orange) and 50 μg (pink) led to significant decreases in cell death compared to 0 μg BEV control (grey) across all timepoints as early as 4 h post-OGD. (**C**) Different dosages of BEVs were applied for 24 h at 24 h pre-OGD (−24 h), and 4 and 24 h post-OGD. Percent cytotoxicity was determined using PI cell viability assay (*N* = 3 slices per group, 4–6 cortical images per slice). Significance was determined by multiple group Kruskal–Wallis testing (ns = not significant, * *p* < 0.05, ** *p* < 0.01, *** *p* < 0.001, *** *p* < 0.0001).

**Figure 4 ijms-23-00620-f004:**
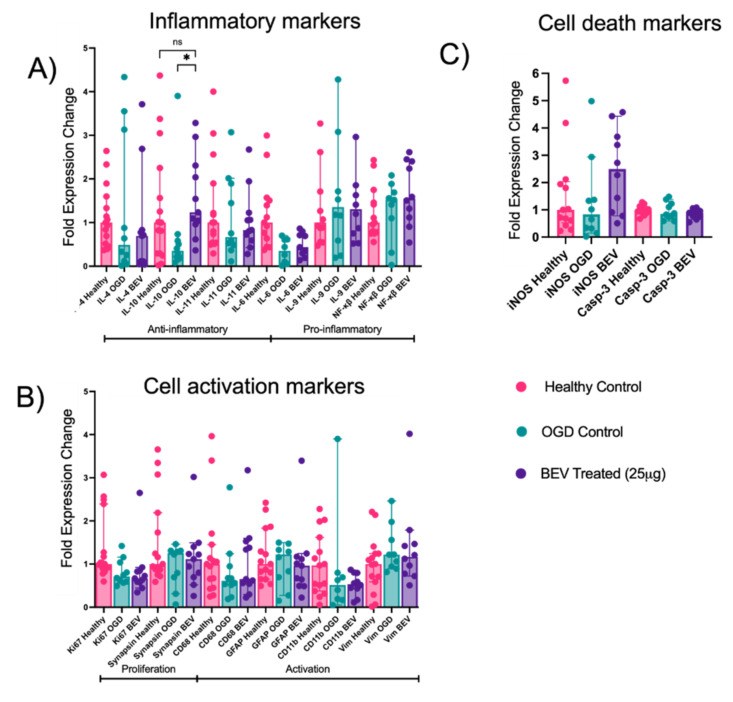
Fold-changes of mRNA markers for (**A**) inflammatory genes, (**B**) cell activation, and (**C**) cell death and stress-related healthy, OGD-conditioned, and BEV-treated ex vivo slices at 24 h exposure (*N* = 7–15 RNA samples (3 replicates per sample); median ± 95% confidence interval; ns: not significant, * = *p* < 0.05).

**Figure 5 ijms-23-00620-f005:**
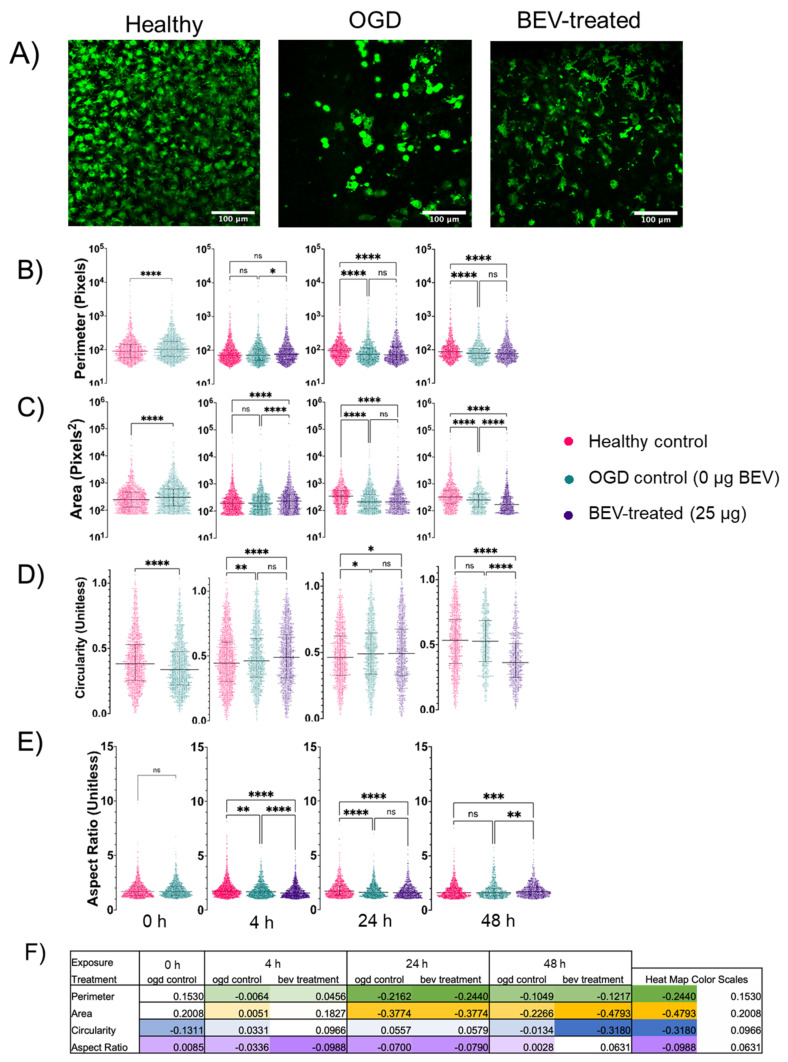
Cell morphology analysis across treatment groups and exposure time. (**A**) Confocal imaging examples of Iba-1-stained microglia at 40× magnification from the cortex of ex vivo brain slices for healthy control, OGD control, and 25 μg BEV-treated slices at an exposure time of 24 h. Scale bars: 100 μm (**B**–**E**) Microglial geometric parameters: (**B**) Perimeter, (**C**) Area Coverage, (**D**) Circularity, (**E**) Aspect Ratio. Graphs display median with interquartile range. ns: not significant, * (*p* < 0.05), ** (*p* < 0.01), *** (*p* < 0.001), and **** (*p* < 0.0001) indicate significant difference with Kruskal–Wallace test adjusted for multiple comparisons. (**F**) Fold change from normal control of each geometric feature. Heat map colors and scales specific to each row and scale are included on the right of the table.

**Figure 6 ijms-23-00620-f006:**
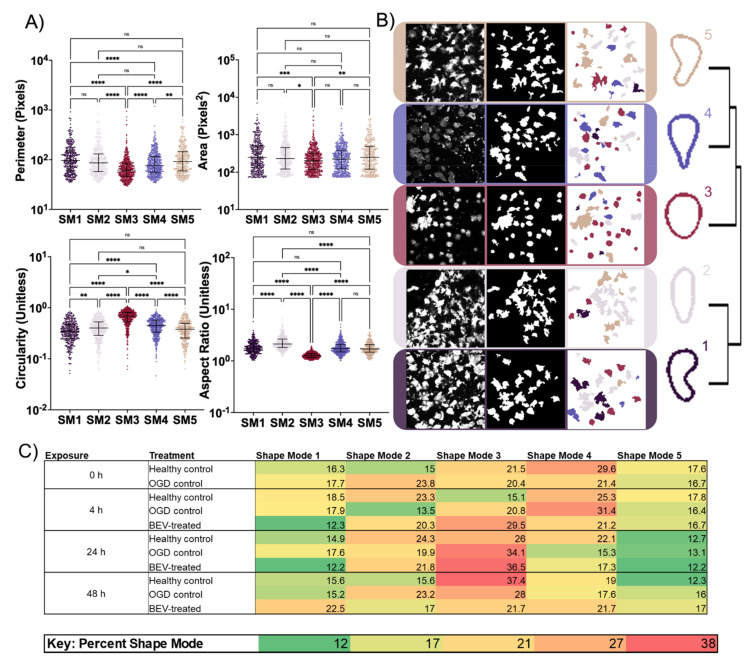
Cell shape mode (SM) parameters compared to geometric features. (**A**) Morphology parameters for the five VAMPIRE SMs. Graphs depict median with interquartile range. ns: not significant, * (*p* < 0.05), ** (*p* < 0.01), *** (*p* < 0.001), and **** (*p* < 0.0001) indicate significant difference with Kruskal–Wallace test adjusted for multiple comparisons. (**B**) Segmentation procedure shown with the original cell image, Li Threshold image, and labeled by color to represent each of the shape modes (colored according to dendrogram on right) after the VAMPIRE method. (**C**) Global heatmaps of percent SM by exposure time and then by treatment.

**Table 1 ijms-23-00620-t001:** Sprague–Dawley Rat mRNA Primer Design for qPCR.

Gene	NCBI Reference Sequence	Forward Primer	Reverse Primer
GAPDH	NM_017008.4	ACTCCCATTCTTCCACCTTTG	ACTCCCATTCTTCCACCTTTG
IL-4	NM_201270.1	GTCACTGACTGTAGAGAGCTATTG	CTGTCGTTACATCCGTGGATAC
IL-6	NM_012589.2	GAAGTTAGAGTCACAGAAGGAGTG	GTTTGCCGAGTAGACCTCATAG
IL-9	NM_001105747.1	GAAGGACGACCCATCATCAAA	ACGGTGTGGTACAATCATCAG
IL-10	NM_012854.2	AGTGGAGCAGGTGAAGAATG	GAGTGTCACGTAGGCTTCTATG
IL-11	NM_133519.5	CTAGCACTTCAAAGGTCCTCAA	ACACCTTGAACCTTGCTATCTC
Ki67	NM_001271366.1	CACACACAAAGAGCCCATAGA	GATTCCTCCTGCCGGTTAAA
Nκβ	NM_001276711.1	GGTTACGGGAGATGTGAAGATG	GTGGATGATGGCTAAGTGTAGG
CD68	NM_001031638.1	CTTGGCTCTCTCATTCCCTTAC	TGTATTCCACTGCCATGTAGTT
Vimentin	NM_031140.1	CTTCCCTGAACCTGAGAGAAAC	GTCTCTGGTTTCAACCGTCTTA
GFAP	NM_017009.2	AAAGACACTGAAACAGGAGAGAG	GGACTGAGCAACCAGGAATAG
Synapsin	NM_001110782.2	GGACGGAAGGGATCACATTATT	ACCACAAGTTCCACGATGAG
CD11b	NM_012711.1	GAGCACCATCTGGGACATAAA	GGCATCAGAGTCCACATCAA
iNOS	NM_012611.3	TGGAGCGAGTTGTGGATTG	CCTCTTGTCTTTGACCCAGTAG
Casp-3	NM_012922.2	GAGCTTGGAACGCTAAGA	CTGACTTGCTCCCATGTAT

## Data Availability

The data can be provided upon request to the corresponding author. Python software is accessible via https://github.com/Nance-Lab, last accessed 22 December 2021.

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
