# Peer review of "Brain Tissue-Derived Extracellular Vesicle Mediated Therapy in the Neonatal Ischemic Brain"

_ijms, 2022, doi:10.3390/ijms23020620_

Round 1

Reviewer 1 Report

Nguyen et al.  reported an interesting study that brain tissue derived extracellular vesicles (BEVs) exert dose- and time-dependent protective effects on ischemic induced injury in brain slice ex vivo. It is demonstrated, by qPCR analysis and microglial morphological changes that, BEV treatment reduced pro-inflammation status, especially lowering IL-10 expression. This study suggests the potential therapeutic role of BEVs for neonatal HIE patients. The followings issues require further clarification.

  1. What is the reason for leaving the sliced brain tissue for 3 days before OGD treatment? Could immune response in the brain tissues be impeded by those 3 days of ex vivo culture?
  2. In figure 3, data points are missing for OCG control (i.e no BEV addition, 0 ug). Or those data point formatted wrongly in white? Therefore, cannot justify the claimed significance
  3. Authors claimed that “BEV dosages follow a trend where increased dosages lead to lower cell cytotoxicity values” (Line 187-188). However, there are no significant between each dosage group. It seems like the lowest dose works just as fine as the highest dose (Again, OGD data points are invisible, cannot justify the claim)
  4. Whole ex vivo ischemic slices were used for RT-qPCR. Is the change of gene expression also related to the architectural region of brain tissues? That may account for the variability in Figure 4?
  5. Regarding OGD conditions, have the authors examined HIF-1a (hypoxia-inducible factor-1a) pathway and its downstream targets for cell survival?
  6. Authors claimed that BEV treatment reduced cytotoxicity in OGD-treated slices. However, the qPCR results showed no difference in cell death markers? Why?
  7. Please clarify the reason of choosing those cell activation markers? They are markers of a mixture of immune cells and mesenchymal cells. Is this set of data an indication of the activation of immune response or proliferation/ survival of brain cells?
  8. What is the relationship between microglial heterogeneity and its function? Authors mentioned this in the discussion. It is better to clarify earlier (for the readers) the significant of analysing different shape mode of microglia, an indicator of the healthiness or functionality of the cells (e.g. inflammatory status etc)?
  9. Related to question #8: why specifically choosing microglial, but not other cell types in the brain tissues? Did the morphology change for other cell types in the brain slides?
  10. What are the contents in BEVs? Can qPCR performed using BEVs to determine the presence of anti-inflammatory markers?
  11. What is the effect of the downstream factors of IL-10 pathway? Do they show consistent trend as IL-10 expression?
  12. How does the therapeutic efficacy of BEV treatment compare to current treatment, i.e. hypothermia?
  13. BEVs are topically applied onto the tissue slices. Presumably, the uptake mechanism of BEVs would be different when administration into patients. What is the route of administration in patient? IV injection? Whether BEV uptake remains time- and dosage-dependent mechanism is unclear? Any in vivo study to support?

Minor comment:

  1. Assumed that the layout in Figure S2 is the same as Figure 2. Please label properly

Reviewer 2 Report

The manuscript provided by authors is very interesting. They investigated the role of BEVs mediating possible therapeutic approaches during brain ischemia. Although well-designed there are some issues that should be addressed by the authors which have been outlined below.

Major:

  • authors should precisely specify what kind of EVs did they use (exosomes, microvesicles, etc). This point is very important to be revealed since different kinds of EVs may induce different responses;
  • authors should explain why is a significant toxicity between healthy controls and 0 ug BEVs (Figure 3)?
  • you performed a dose-dependent evaluation only for 0, 5, 12.5, 25, 50 ug, what about the lower doses than 5 ug BEV? Microglia are know to be highly sensitive to very low-doses of different agents, especially pathogenic stressors (check Lajqi et al. 2019) - please discuss this issue;
  • authors should reveal any mechanistic alterations that promote these effects arising from BEVs (i.e., signaling proteins).;
  • authors should pay attention to the data of microglia staining by IBA-1 since the IBA-1 antigen is not specific for microglia in vivo, it rather stains all myeloid-derived cells (resident microglia and migrating peripheral monocytes/macrophages). After brain injury, both cell populations are involved in inflammatory reactions.

Minor:

  • authors should report the stability of their BEVs (for example measuring the Zeta potential);
  • please provide the purity of your BEVs (% of 30-200 nm EVs);
  • the authors should provide us the original TEM pictures as supplementary data and also show any TEM pictures where we see quantitatively the number of EVs (zoom out);
  • please check for typing errors.

Round 2

Reviewer 2 Report

Thanks for improving the manuscript and answering my concerns.